# VLM-PTA: Exploiting Page Table Attack to Deplete the Intelligence of VLMs

## Abstract

Vision language models (VLMs) excel at multimodal tasks such as image captioning and visual question answering, yet they remain vulnerable to input manipulation attacks (e.g., jailbreak and adversarial attacks). However, the vulnerability of VLMs to adversarial weight perturbation remains largely underexplored. Our initial investigation reveals that VLMs remain extremely resilient to conventional weight corruption attacks leveraging memory fault injections (e.g., bit-flip attacks). As a consequence, we propose the first successful adversarial weight perturbation attack against VLMs *(VLM-PTA)*. Our attack leverages *page table attack* (PTA), a well-established memory fault injection technique. In the main memory, each weight block consists of a group of weights located at a specific address. Consequently, a bit-flip in the page frame number replaces a *victim weight block* of a VLM with another *substitute weight block*. However, the algorithmic challenge in creating a formal attack is that the random injection of weight replacement into the model fails to cause any detrimental impact on the model's performance. Therefore, we theoretically analyze the bottleneck of the PTA-based fault injection mechanism and propose a novel estimation method (Block-Flip) to maximize attack effectiveness and efficiency. VLM-PTA is the most successful weight perturbation attack against VLMs optimized to achieve adversarial objectives with an extremely low overhead, bypassing existing defenses.

## 1 Introduction

In recent years, vision language models (VLMs) have achieved significant advancements in interpreting and reasoning across visual and textual modalities, achieving state-of-the-art performance in a wide range of multimodal tasks, such as image captioning (Fei et al., 2023; Ramos et al., 2023), visual question answering (Chen et al., 2022; Nguyen et al., 2024), and cross-modal retrieval (Chen et al., 2023). Despite these impressive capabilities, recent studies have revealed that VLMs remain vulnerable to a wide range of input manipulation attacks, such as jailbreak attacks (Shayegani et al., 2023; Niu et al., 2024; Qi et al., 2024) and adversarial attacks (Cui et al., 2024; Zhao et al., 2023; Tu et al., 2024). However, models' internal parameter perturbation, formally known as adversarial weight attack( (Yao et al., 2020; Dong et al., 2023; Li et al., 2024; Lin et al., 2025; Chen et al., 2021; Ahmed et al., 2024; Rakin et al., 2020)) has not been investigated against VLMs.

Adversarial weight perturbation can be broadly classified into two categories: first, backdoor or Trojan attack, which also requires external data manipulation (Ahmed et al., 2024; Chen et al., 2021; Rakin et al., 2020); and second, perturbing the weights only through remote memory fault injection to achieve attacker-designed model behavior without corrupting the input at all (Yao et al., 2020; Dong et al., 2023; Lin et al., 2025). Our work focuses on the second category, which aims to leverage memory fault injection through remote side-channels such as rowhammer exploitation (Kim et al., 2014). Prior works have shown (Yao et al., 2020; Dong et al., 2023; Lin et al., 2025) that leveraging such a unique attack vector, the intelligence of modern deep learning models

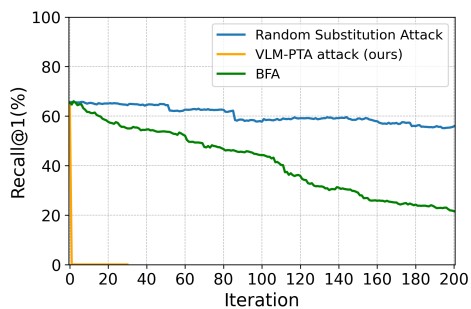

Figure 1: Random weight substitution and BFA fails for CLIP, while VLM-PTA depletes the accuracy $\approx 0$ with $< 5$ iterations.

can be destroyed by only a limited amount of bit-flips (e.g., 1-20 bits) in the quantized model weights. In contrast, our observation shows that after applying the conventional bit-flip attacks (BFA) in VLMs, the model performance still maintains at 21.61 after 200 bit-flips, as shown in Figure 1.

This observation led our investigation of alternative memory fault injection mechanism that can potentially exploit the vulnerability of VLMs more effectively. In this work, we propose a powerful alternative, which is a well-established memory fault injection mechanism demonstrated in many prior works (Frigo et al., 2020; Jattke et al., 2022; Gruss et al., 2016; Seaborn & Dullien, 2015; Van Der Veen et al., 2016; Xiao et al., 2016; Zhang et al., 2020; Ahmed et al., 2024) known as the page table attack. In the page table attack, the attacker uses the rowhammer (Kim et al., 2014) to induce a bit flip in the Page Frame Number (PFN) to compromise memory systems. By flipping bits in the PFN, the attacker can thus replace any *victim block W1* with a *substitute block W2* as shown in Figure 2. However, when we apply the above weight replacement mechanism randomly to replace any victim block of weights in VLMs, it fails (shown in Figure 1). As a result, to develop the first successful weight perturbation attack against VLM, our work contributes three research outcomes:

- We propose a novel attack algorithm VLM-PTA that can jointly optimize the search process of the victim and the substitute weight block to minimize attack overhead (e.g., energy, timing). The proposed search algorithm is designed to achieve two distinct adversarial goals for VLMs, which perform both retrieval and generative captioning: i) untargeted attack to disrupt the caption generation process and ii) targeted attack to retrieve a target caption consistently.

- The failure of prior weight perturbation attacks on VLM lacks explanation, as they are often designed heuristically without any theoretical foundation behind their design principle. In contrast, our work is the first weight perturbation technique to provide a rigorous theoretical analysis underpinning attack design choices and to offer an explanation for its effectiveness.

- Finally, we provide an extensive experimental analysis that demonstrates the effectiveness of VLM-PTA in executing targeted and untargeted attacks across different model architectures, tasks, and datasets. VLM-PTA is the most successful adversarial weight attack on VLMs evaluated in terms of highest attack effectiveness, lowest attack overhead and ability to break existing defenses.

## 2 BACKGROUND AND RELATED WORKS

**Vision Language Models (VLMs).** The general structure of VLMs consists typically of two encoders: visual encoder and textual encoder. The visual encoder is commonly implemented using Convolutional Neural Networks (CNNs) or vision transformers (ViTs), while the textual encoder is generally based on language models. The primary purpose of these encoders is to project their respective modalities into a shared latent space, where multimodal interactions are captured through mechanisms such as attention (Lu et al., 2019) or contrastive learning objectives (Jia et al., 2021).

**Attacks on VLMs.** Despite the remarkable capabilities of VLMs in processing and aligning text–image information, a growing body of research has highlighted their vulnerabilities to various types of attacks. Jailbreaking is one type of attack, in which adversaries attempt to bypass the model safety mechanisms to elicit harmful or restricted content, such as generating toxic or unsafe outputs (Shayegani et al., 2023; Qi et al., 2024). Adversarial attacks are another well studied category of attacks on VLMs, where carefully crafted perturbations are introduced into the input image or text to mislead the model outputs (Cui et al., 2024; Zhao et al., 2023; Tu et al., 2024).

## 3 THREAT MODEL

Our attack adopts a standard practical threat model following the attacker privileges established by previous adversarial weight attacks (Frigo et al., 2020; Jattke et al., 2022; Gruss et al., 2016; Seaborn & Dullien, 2015; Van Der Veen et al., 2016; Xiao et al., 2016; Zhang et al., 2020; Ahmed et al., 2024) exploitations. The attack requires specific system-level privileges that allows them to reverse-engineer the memory addressing scheme of models weights and flip bits in the page frame number. Several side-channel attacks have demonstrated the feasibility of such attacks in main memory of a DRAM (Pessl et al., 2016; Yao et al., 2020; Hu et al., 2020; Lin et al., 2025; Yan et al., 2020; Xiang et al., 2020; Yu et al., 2020; Rakin et al., 2022). Additionally, by following the standard practice of conventional white-box attack, we assume the attacker can access to model

weights, architecture, and a batch of test data, but does not require training data or hyperparameter. This white-box assumption can be utilized through remote side channel attacks (Yan et al., 2020; Xiang et al., 2020; Yu et al., 2020; Rakin et al., 2022) across different platforms. In summary, we adopt the standard practical threat model assumption following the convention of existing adversarial input (Madry et al., 2018) and weight attacks (Lin et al., 2025; Ahmed et al., 2024; Yao et al., 2020; Dong et al., 2023). A detailed explanation of threat model is on the Appendix C.

## 4 PROPOSED ATTACK: VLM-PTA

We propose *VLM-PTA*, a page-table attack designed to perform the first adversarial weight perturbation on Vision Language Models (VLMs). Our attack is conducted at runtime by injecting faults into the page tables of memory addresses (Zhang et al. (2020)). Since random fault injection fails as shown in Figure 1, our attack is supported by a weight block searching algorithm *(Block-Flip)* whose goal is to answer what part of the VLM weight block should be perturbed and by how much. The attacker performs the search step offline using a second copy of the VLM to record these memory addresses of the vulnerable weight block. At runtime, the attacker launches the attack on the recorded address set, which has already

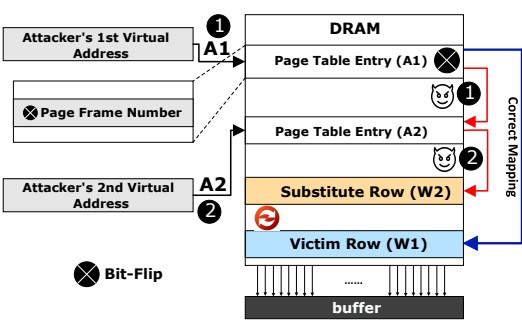

Figure 2: Overview of using rowhammer to perform fault injection on Page Table Entry (PTE). In normal execution, PTE $A1$ maps to Victim Row ($W1$ weight block). On VLM-PTA: (1) attacker hammers $A1$ to redirect it to PTE $A2$, (2) PTE $A2$ is mapped to Substitute Row $W_2$.

been optimized to minimize the attack overhead, effectively achieving the attacker's objectives.

### 4.1 FAULT INJECTION OVERVIEW.

Page tables are components of virtual memory systems that translate virtual addresses used by processes into corresponding physical addresses in RAM. Figure 2 shows that in ordinary execution of a program, the virtual memory row (A1) should have been correctly mapped to its corresponding physical address (includes W1 weight block). However, in VLM-PTA, the attacker employs double-sided rowhammer to flip bits in the first Page Frame Number (PFN) within their Page Table Entries (PTE) in step ❶, causing a particular PTE (A1) to point to the second-page table address (A2) instead. This manipulation grants the attacker read or write access to PTE (A2), which allows the attacker to manipulate the pointers to any desired physical pages, such as the Substitute Row (includes W2) in step ❷. Hence, the attacker successfully replaced W1 with the W2 weight block.

### 4.2 MATHEMATICAL MODELING OF THE FAULT.

To develop the attack algorithm, we mathematically model the above fault injection technique. Consider a VLM denoted as $f(\cdot)$, with its weights stored in a memory block. We define a set of virtual memory addresses as $\mathcal{A} = \{a_1, a_2, \ldots, a_n\}$, where each address $a_i$ is a 32-bit value pointing to a physical address containing weight block $\mathbf{w}_i$. Each weight block $\mathbf{w}_i$ contains 1024 weights used in the VLM. Therefore, we have a set of weight blocks represented by $\mathcal{W} = \{\mathbf{w}_1, \mathbf{w}_2, \ldots, \mathbf{w}_n\}$, which collectively hold the weights of the VLM. Our attack flips bits in the page frame number of the memory address $a_i$, resulting in the replacement of a *victim weight block* $\mathbf{w}_i$ at $a_i$ with a new weight block $\mathbf{w}_j$ taken from a different memory address $a_j$, referred as *substitute weight block*.

### 4.3 VLM-PTA OBJECTIVES.

We propose two variants of the attack algorithm as outlined below:

*1) VLM-PTA Untargeted (VLM-PTA-U):* In this goal, VLM-PTA causes the VLM to generate unrelated or retrieve the least related caption ($\mathbf{y}$) for all input images $\mathbf{x} \in \mathcal{X}$.

$$\max_{\hat{\mathcal{W}}} \mathbb{E}_{\mathbf{x} \sim \mathcal{X}} \left[ \mathcal{L}(f(\mathbf{x}, \hat{\mathcal{W}}), \mathbf{y}) \right] \tag{1}$$

*2)VLM-PTA Targeted (VLM-PTA-T).* Under this objective, attack causes the VLM to retrieve the target caption ($\mathbf{y}_t$) for all input images $\mathbf{x} \in \mathcal{X}$.

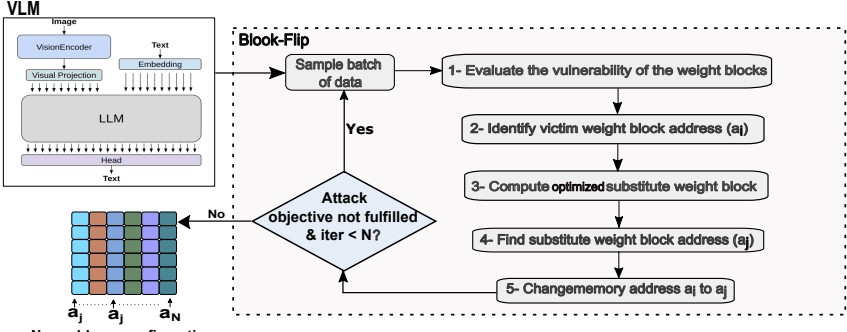

Figure 3: Overview of Block-Flip algorithm.

$$\min_{\hat{\mathcal{W}}} \mathbb{E}_{\mathbf{x} \sim \mathcal{X}} \left[ \mathcal{L}(f(\mathbf{x}, \hat{\mathcal{W}}), \mathbf{y}_t) \right] \tag{2}$$

In both 1 and 2, the quantification of loss $\mathcal{L}(\cdot, \cdot)$ depends on the VLM task: for retrieval models (e.g., CLIP (Radford et al., 2021)), loss corresponds to an image–text similarity such as negative cosine similarity, for generative captioning models (e.g., BLIP (Li et al., 2022)), loss corresponds to token-level cross entropy between generated and ground-truth captions. By perturbing the VLM's weights from $\mathcal{W}$ to $\hat{\mathcal{W}}$, the attacker seeks to maximize or minimize loss, thus achieving the aforementioned attack objectives.

In addition to the primary objectives, we design VLM-PTA to maintain two additional optimization constraints: (1) minimize the number of attack iterations to reduce overhead, i.e, the amount of weight blocks being modified and (2) restrict the weight modification within the existing set of weight blocks to avoid unintended memory faults or reduce any additional write operation following the practice of prior page table attacks (Zhang et al. (2020); Ahmed et al. (2024)). Specifically, a victim weight block $\mathbf{w}_i$ is replaced with an optimized block $\mathbf{w}_j \in \mathcal{W}$ by altering the address from $a_i$ to $a_j \in \mathcal{A}$. As a result, our attack ensures that the altered addresses $\hat{\mathcal{A}}$ and modified set of weight blocks $\hat{\mathcal{W}}$ are confined within the initial weight and address set, i.e., $\hat{\mathcal{A}} \subset \mathcal{A}$ and $\hat{\mathcal{W}} \subset \mathcal{W}$ and at the same time minimizing the amount of weight alteration to reduce attack cost. Incorporating these constraints, we redefine the attack objectives in equation 1 and equation 2 as follows:

$$\max_{\hat{\mathcal{W}}} \mathbb{E}_{\mathbf{x} \sim \mathcal{X}} \left[ \mathcal{L}(f(\mathbf{x}, \hat{\mathcal{W}}), \mathbf{y}) \right], \quad \text{s.t.} \quad \mathcal{D}(\hat{\mathcal{A}}, \mathcal{A}) \leq \gamma_u, \quad \hat{\mathcal{A}} \subset \mathcal{A}, \quad \hat{\mathcal{W}} \subset \mathcal{W} \tag{3}$$

$$\min_{\hat{\mathcal{W}}} \mathbb{E}_{\mathbf{x} \sim \mathcal{X}} \left[ \mathcal{L}(f(\mathbf{x}, \hat{\mathcal{W}}), \mathbf{y}_t) \right], \quad \text{s.t.} \quad \mathcal{D}(\hat{\mathcal{A}}, \mathcal{A}) \leq \gamma_t, \quad \hat{\mathcal{A}} \subset \mathcal{A}, \quad \hat{\mathcal{W}} \subset \mathcal{W} \tag{4}$$

Where $\hat{\mathcal{A}}$ represents the set of new addresses due to memory address fault injection, $\hat{\mathcal{W}}$ represents the set of new weights resulting from the weight replacement. The function $\mathcal{D}(\cdot, \cdot)$ denotes the Hamming distance between the unaltered weight addresses $\mathcal{A}$ and the altered weight addresses $\hat{\mathcal{A}}$, and $\gamma_u$ is our maximum budget of bit-flips to alter the addresses.

### 4.4 PROPOSED (BLOCK-FLIP) ALGORITHM

The proposed Block-Flip algorithm identifies the vulnerable victim and its corresponding substitute weight block progressively, one at a time (attack iteration). To achieve the attacker-defined objectives on 3 and 4, we devise a five-step process for each attack iteration, as shown in Figure 3, and each specific attack design choices are supported by theoretical analysis.

**First Step (Evaluate the vulnerability of the weight blocks):** In this step, the weight blocks are evaluated according to their impact in achieving the attack goals using loss gradients. Consequently, blocks with the highest gradients are most vulnerable to the weight perturbation. The gradient of the $i^{th}$ weight block $\mathbf{w}_i$ as follows:

$$\mathbf{g}_i = \begin{bmatrix} \frac{\partial \mathcal{L}(\mathcal{W})}{\partial \mathbf{w}_{i1}} & \cdots & \frac{\partial \mathcal{L}(\mathcal{W})}{\partial \mathbf{w}_{i128}} \end{bmatrix}^T \tag{5}$$

**Second Step (Identify the address of the victim weight block):** After obtaining the vulnerability of each weight block using equation 5, attacker must choose only one weight block $\mathbf{w}_t$ as victim block on each iteration. So, we define a rank metric of each weight block $\mathbf{w}_i$ as the l2-norm of its gradient vector in equation 5, i.e.,

$$\text{rank}(\mathbf{w}_i) = \|\mathbf{g}_i\| \tag{6}$$

Using the rank metric in equation 6, we select the highest-ranked weight block $\mathbf{w}_i$, indicating that modifying this $i^{th}$ weight block will have a larger impact on the attack goal.

$$\mathbf{w}_t = \underset{\mathbf{w}_i \in \mathcal{W}}{\text{argmax}} \; \text{rank}(\mathbf{w}_i) \tag{7}$$

**Third Step (Compute the optimized substitute weight block):** After identifying the victim block, the attacker must decide how to modify $\mathbf{w}_t$ to achieve the attack goal defined in equation 4 and 3. To maintain the constraint of these equations, the next step in the search process must answer two questions:

- **Q1:** *Can we derive an optimized substitute block for the identified victim block $\mathbf{w}_t$ in the previous step to maximize the attack goal?*
- **Q2:** *Does this optimized substitute block exists withing the set of VLM weight blocks?*

In the following Lemmas, we show that there is a substitute block $\mathbf{w}_r^*$ that maximizes the effectiveness of the attack, and under additional assumptions (Neal, 2012; Matthews et al., 2018; de G. Matthews et al., 2018), that $\mathbf{w}_r^*$ lies within the VLM.

**Lemma 4.1** (Optimized Substitute Weight Block). *Consider a VLM with a victim weight block $\mathbf{w}_t \in \mathbb{R}^d$ and a corresponding differentiable loss function $\mathcal{L}(\mathcal{W})$. Assume that $\|\nabla_{\mathbf{w}_t} \mathcal{L}(\mathcal{W})\| > 0$. Then for any such victim weight block $\mathbf{w}_t$, there exists a substitute weight block $\mathbf{w}_r^*$ such that $\mathcal{L}(\mathcal{W}') > \mathcal{L}(\mathcal{W})$ for a positive learning rate $\alpha$, where $\mathbf{w}_r^* = \mathbf{w}_t + \alpha \nabla_{\mathbf{w}_t} \mathcal{L}(\mathcal{W})$ and $\mathcal{W}' = \mathcal{W} \setminus \mathbf{w}_t \cup \mathbf{w}_r^*$.*

**Proof.** Let $\mathcal{L}(\mathcal{W})$ be the loss function of VLM, which is differentiable with respect to the weights $\mathcal{W}$. The gradient of the loss function w.r.t. $\mathbf{w}_t$ is given by:

$$\nabla_{\mathbf{w}_t} \mathcal{L}(\mathcal{W}) = \left( \frac{\partial \mathcal{L}(\mathcal{W})}{\partial \mathbf{w}_{t1}}, \frac{\partial \mathcal{L}(\mathcal{W})}{\partial \mathbf{w}_{t2}}, \dots, \frac{\partial \mathcal{L}(\mathcal{W})}{\partial \mathbf{w}_{td}} \right)$$

Consider a substitute weight block $\mathbf{w}_r^*$ defined as: $\mathbf{w}_r^* = \mathbf{w}_t + \alpha \nabla_{\mathbf{w}_t} \mathcal{L}(\mathcal{W})$, where $\alpha$ is a positive learning rate. Using a first-order Taylor expansion of $\mathcal{L}(\mathcal{W}')$ around $\mathbf{w}_t$, we have[1]:

$$\mathcal{L}(\mathcal{W}') = \mathcal{L}(\mathcal{W} - \mathbf{w}_t + \mathbf{w}_r^*) \approx \mathcal{L}(\mathcal{W}) + \nabla_{\mathbf{w}_t} \mathcal{L}(\mathcal{W})^T (\mathbf{w}_r^* - \mathbf{w}_t) \tag{8}$$

Substituting $\mathbf{w}_r^* = \mathbf{w}_t + \alpha \nabla_{\mathbf{w}_t} \mathcal{L}(\mathcal{W})$:     $\mathcal{L}(\mathcal{W}') \approx \mathcal{L}(\mathcal{W}) + \nabla_{\mathbf{w}_t} \mathcal{L}(\mathcal{W})^T \left( \alpha \nabla_{\mathbf{w}_t} \mathcal{L}(\mathcal{W}) \right)$

Simplifying, we get:   $\mathcal{L}(\mathcal{W}') \approx \mathcal{L}(\mathcal{W}) + \alpha \|\nabla_{\mathbf{w}_t} \mathcal{L}(\mathcal{W})\|^2$

Since $\alpha > 0$ and $\|\nabla_{\mathbf{w}_t} \mathcal{L}(\mathcal{W})\| > 0$, we have:   $\mathcal{L}(\mathcal{W}') > \mathcal{L}(\mathcal{W})$

Hence, this shows a strict increase in the loss function. This Lemma shows the existence of a substitute weight block for loss maximization 3. Similarly, the substitute weight block for the loss minimization problem (i.e., VLM-PTA-T in 4) can be derived as $\mathcal{L}(\mathcal{W}') < \mathcal{L}(\mathcal{W})$ when $\mathbf{w}_r^* = \mathbf{w}_t - \alpha \nabla_{\mathbf{w}_t} \mathcal{L}(\mathcal{W})$. $\qquad \square$

**Implication of Lemma 4.1 (Q1.Answer):** *It shows that for each victim weight block $\mathbf{w}_t$, there exists a substitute weight block $\mathbf{w}_r^*$ that maximizes the attack Objective for the current attack iteration.*

**Lemma 4.2** (Existence of Optimized Substitute Weight Block in VLM). *Consider a VLM with weight block set $\mathcal{W} = \{\mathbf{w}_1, \mathbf{w}_2, \dots, \mathbf{w}_n\}$ where each $\mathbf{w}_i \in \mathbb{R}^d$ is a d-dimensional vector of i.i.d normal random variables with zero mean and variance $\sigma^2$, i.e., $\mathbf{w}_i = (\mathbf{w}_{i1}, \mathbf{w}_{i2}, \dots, \mathbf{w}_{id})$ where $\mathbf{w}_{ij} \sim N(0, \sigma^2)$ for all $i$ and $j$. Let the optimized substitute weight block $\mathbf{w}_r^* \in \mathbb{R}^d$ be a d-dimensional block vector with weights $\mathbf{w}_r^* = (\mathbf{w}_{r1}^*, \mathbf{w}_{r2}^*, \dots, \mathbf{w}_{rd}^*)$, where $\mathbf{w}_{rj}^* \sim N(0, \sigma^2)$ for all $j$. Then, the probability $P(\mathbf{w}_r^* \in \mathcal{W}) \to 1$ as $n \to \infty$.*

---

[1]In equation 8, we slightly abuse notation by expressing $\mathcal{W} \setminus \mathbf{w}_t \cup \mathbf{w}_r^*$ as $\mathcal{W} - \mathbf{w}_t + \mathbf{w}_r^*$, since $\mathcal{L}(\mathcal{W}')$ is expanded only around $\mathbf{w}_t$.

**Proof.** We consider a probabilistic approach in terms of proximity within an $\epsilon$-neighborhood. The probability that $\mathbf{w}_r^*$ is within an $\epsilon$-neighborhood of $\mathbf{w}_i$ is $P(\|\mathbf{w}_i - \mathbf{w}_r^*\| \leq \epsilon)$, where $\|\cdot\|$ denotes the Euclidean norm. The difference $\mathbf{w}_i - \mathbf{w}_r^*$ is a $d$-dimensional vector where each element $(\mathbf{w}_{ij} - \mathbf{w}_{rj}^*) \sim N(0, 2\sigma^2)$. Then the norm $\|\mathbf{w}_i - \mathbf{w}_r^*\|$ follows a chi distribution with $d$ degrees of freedom and a scaling factor of $\sigma\sqrt{2}$, i.e., $\frac{\|\mathbf{w}_i - \mathbf{w}_r^*\|}{\sigma\sqrt{2}} \sim \chi_d$. The probability that $\|\mathbf{w}_i - \mathbf{w}_r^*\| \leq \epsilon$ is as follows:

$$P(\|\mathbf{w}_i - \mathbf{w}_r^*\| \leq \epsilon) = P\left(\chi_d \leq \frac{\epsilon}{\sigma\sqrt{2}}\right) = F_{\chi_d}\left(\frac{\epsilon}{\sigma\sqrt{2}}\right)$$

where $F_{\chi_d}$ is the CDF of the chi distribution with $d$ degrees of freedom.

For $n$ weight block vectors, the probability that none of the $n$ block vectors is within $\epsilon$ of $\mathbf{w}_r^*$ is:

$$P\left(\bigcap_{i=1}^n \{\|\mathbf{w}_i - \mathbf{w}_r^*\| > \epsilon\}\right) = \left[1 - F_{\chi_d}\left(\frac{\epsilon}{\sigma\sqrt{2}}\right)\right]^n$$

Therefore, the probability that at least one vector $\mathbf{w}_i$ is within $\epsilon$ of $\mathbf{w}_r^*$ is:

$$P\left(\bigcup_{i=1}^n \{\|\mathbf{w}_i - \mathbf{w}_r^*\| \leq \epsilon\}\right) = 1 - \left[1 - F_{\chi_d}\left(\frac{\epsilon}{\sigma\sqrt{2}}\right)\right]^n$$

As $n \to \infty$, the term $\left[1 - F_{\chi_d}\left(\frac{\epsilon}{\sigma\sqrt{2}}\right)\right]^n$ approaches 0, with $\epsilon > 0$. Hence, $1 - \left[1 - F_{\chi_d}\left(\frac{\epsilon}{\sigma\sqrt{2}}\right)\right]^n$ approaches to 1. Hence proving the statement of the lemma. $\square$

**Implication of Lemma 4.2 (Q2.Answer):** *Lemma 4.2 shows that with an increasing number of weight blocks $n$, the probability that at least one block $\mathbf{w}_i$ is within $\epsilon$ of the optimized substitute block $\mathbf{w}_r^*$ approaches 1. Indicating this probability is nearly one even for a small-scale model with 0.2 million parameters ($n \approx 2000$) when $\epsilon = 0.05$ and $\sigma = 1$. Considering the scale of the VLM text and image encoder, this conclusion holds.*

Once we theoretically analyze that there exists an optimized substitute block for every vulnerable victim block, the next step is to design a strategy to find this optimized substitute weight block $\mathbf{w}_r^*$ for the victim weight block $\mathbf{w}_t$ (identified in the previous step 3).

**Strategy to calculate optimized substitute weight block.** Finding the optimized substitute block requires two levels of optimization: *First*, which direction to change, and *Second,* how much to change the victim weight block to achieve the objectives defined in equation 3 and 4 faster. The first part can be optimized using the principle of gradient descent (targeted) or ascent (un-targeted) using the following update equations: $\mathbf{w}_r^* = \mathbf{w}_t - \hat{\alpha} \cdot \mathbf{g}_t$ and $\mathbf{w}_r^* = \mathbf{w}_t + \hat{\alpha} \cdot \mathbf{g}_t$ respectively. However, the second part (i.e., speed of attack convergence) depends on choosing an optimized learning rate; next, Theorem 4.3 derives an optimized learning rate.

**Theorem 4.3 (Optimized Learning Rate).** *Consider a VLM with a weight block set $\mathcal{W} = \{\mathbf{w}_1, \mathbf{w}_2, \ldots, \mathbf{w}_n\}$, any victim weight block $\mathbf{w}_t \in \mathcal{W}$ and the optimized substitute weight block $\mathbf{w}_r^*$, where each $\mathbf{w}_i$ and $\mathbf{w}_r^*$ is a $d$-dimensional vector of i.i.d normal random variables with zero mean and variance $\sigma^2$. For any victim weight block $\mathbf{w}_t$ and a corresponding differentiable loss function $\mathcal{L}(\mathcal{W})$, the estimate $\hat{\alpha}$ is an unbiased estimate of the optimized learning rate $\alpha$ to reach the desired substitute weight block $\mathbf{w}_r^*$ given by:*

$$\hat{\alpha} = \frac{1}{n-1} \frac{\sum_{i \neq t} \|\mathbf{w}_t - \mathbf{w}_i\|}{\|\nabla_{\mathbf{w}_t} \mathcal{L}(\mathcal{W})\|} \tag{9}$$

**Proof.** By Lemma 4.1 and 4.2, there exists a substitute weight block $\mathbf{w}_r^*$ within VLM weight block set that minimizes the loss $\mathcal{L}(\cdot)$. The gradient descent step to reach the optimized substitute weight block is given by: $\mathbf{w}_r^* = \mathbf{w}_t + \alpha \nabla_{\mathbf{w}_t} \mathcal{L}(\mathcal{W})$, where $\alpha$ is the optimized learning rate. Simplifying, we get:

$$\alpha = \frac{\|\mathbf{w}_t - \mathbf{w}_r^*\|}{\|\nabla_{\mathbf{w}_t} \mathcal{L}(\mathcal{W})\|} \tag{10}$$

Now, to prove the statement of the Theorem, we need to show that $\hat{\alpha}$ in equation 9 is an unbiased estimate of the optimized learning rate given in equation 10. The expected value of $\hat{\alpha}$ is given by:

$$\mathbb{E}[\hat{\alpha}] = \mathbb{E}\left[\frac{1}{n-1} \frac{\sum_{i \neq t} \|\mathbf{w}_t - \mathbf{w}_i\|}{\|\nabla_{\mathbf{w}_t} \mathcal{L}(\mathcal{W})\|}\right] = \frac{1}{n-1} \mathbb{E}\left[\frac{\sum_{i \neq t} \|\mathbf{w}_t - \mathbf{w}_i\|}{\|\nabla_{\mathbf{w}_t} \mathcal{L}(\mathcal{W})\|}\right] = \frac{1}{n-1} \frac{\sum_{i \neq t} \mathbb{E}[\|\mathbf{w}_t - \mathbf{w}_i\|]}{\|\nabla_{\mathbf{w}_t} \mathcal{L}(\mathcal{W})\|}$$

Since $\mathbf{w}_{ij}$ are i.i.d normal random variables with zero mean and variance $\sigma^2$ for all $i \in [1, n]$ and $j \in [1, d]$, the difference $\mathbf{w}_t - \mathbf{w}_i$ is also normally distributed with zero mean and variance $2\sigma^2$. And it can be shown that the norm of this difference $\|\mathbf{w}_t - \mathbf{w}_i\|$ follows a chi distribution with $d$ degrees of freedom, scaled by $\sqrt{2\sigma^2}$. The expected value of the norm of a chi-distributed variable with $d$ degrees of freedom is given by: $\mathbb{E}[\|\mathbf{w}_t - \mathbf{w}_i\|] = \sqrt{2\sigma^2} \cdot \frac{\Gamma\left(\frac{d+1}{2}\right)}{\Gamma\left(\frac{d}{2}\right)}$

Therefore, the expected value of $\hat{\alpha}$ is:

$$\mathbb{E}[\hat{\alpha}] = \frac{1}{n-1} \frac{\sum_{i \neq t} \sqrt{2\sigma^2} \cdot \frac{\Gamma\left(\frac{d+1}{2}\right)}{\Gamma\left(\frac{d}{2}\right)}}{\|\nabla_{\mathbf{w}_t}\mathcal{L}(\mathcal{W})\|} = \frac{(n-1)\sqrt{2\sigma^2} \cdot \frac{\Gamma\left(\frac{d+1}{2}\right)}{\Gamma\left(\frac{d}{2}\right)}}{(n-1)\|\nabla_{\mathbf{w}_t}\mathcal{L}(\mathcal{W})\|} = \frac{\sqrt{2\sigma^2} \cdot \frac{\Gamma\left(\frac{d+1}{2}\right)}{\Gamma\left(\frac{d}{2}\right)}}{\|\nabla_{\mathbf{w}_t}\mathcal{L}(\mathcal{W})\|} \quad (11)$$

By Lemma 4.2, $\exists \mathbf{w}_r^*$ in a VLM. Since, each $\mathbf{w}_{rj} \sim \mathcal{N}(0, \sigma^2), \forall j \in [1, d]$, it can be shown that $\|\mathbf{w}_t - \mathbf{w}_r^*\|$ is also a chi-distributed variable with $d$ degrees of freedom where the expectation is given by:

$$\mathbb{E}[\|\mathbf{w}_t - \mathbf{w}_r^*\|] = \sqrt{2\sigma^2} \cdot \frac{\Gamma\left(\frac{d+1}{2}\right)}{\Gamma\left(\frac{d}{2}\right)}$$

Therefore, the expected value of $\alpha$ is also given by equation 11. Hence, we have shown that $\hat{\alpha}$ is an unbiased estimate of the optimized learning rate $\alpha$. $\qquad\square$

Using equation 9, calculating the optimized substitute weight block for VLM-PTA-U is:

$$\mathbf{w}_r^* = \mathbf{w}_t + \hat{\alpha} \cdot \mathbf{g}_t = \mathbf{w}_t + \frac{1}{n-1} \sum_{i=1, i \neq t}^{n} \|\mathbf{w}_t - \mathbf{w}_i\| \cdot \frac{\mathbf{g}_t}{\|\mathbf{g}_t\|} \quad (12)$$

However, calculating the optimized substitute weight block for VLM-PTA-T is:

$$\mathbf{w}_r^* = \mathbf{w}_t - \hat{\alpha} \cdot \mathbf{g}_t = \mathbf{w}_t - \frac{1}{n-1} \sum_{i=1, i \neq t}^{n} \|\mathbf{w}_t - \mathbf{w}_i\| \cdot \frac{\mathbf{g}_t}{\|\mathbf{g}_t\|}$$

Figure 4 shows that using optimized learning rate to calculate the optimized substitute weight block improves the convergence of the attack by $10\times$.

**Fourth Step (Find the address of the optimized substitute weight block):** On this step, we first identify the set of top-k similar weight blocks to our optimized substitute weight block $\mathbf{w}_r^*$ using the dot product as a similarity metric: $\mathcal{W}_r = \left\{ \mathbf{w}_i \mid \mathbf{w}_i \in \mathcal{W}, \mathbf{w}_i \neq \mathbf{w}_t \text{ and } \mathbf{w}_i \in \text{top-k}(\mathbf{w}_i^T \mathbf{w}_r^*) \right\}$

Next, among the above candidates of the substitute weight block, we compute the Hamming distance between the address of the victim weight block $\mathbf{w}_t$ and each of the weight blocks in set $\mathcal{W}_r$. Then we select the weight block with the minimum Hamming distance as the substitute weight block $\mathbf{w}_r$ to minimize attack overhead even further:

$$\mathbf{w}_r = \underset{\mathbf{w}_i \in \mathcal{W}_r}{\arg\min} \mathcal{D}(a_t, a_i) \quad (13)$$

Figure 4: Example of attack convergence showing that scaled gradients $(\hat{\alpha} \cdot \mathbf{g}_t)$ achieve faster convergence.

where $a_i$ is address of weight block $\mathbf{w}_i$ and $a_t$ is the address of victim weight block $\mathbf{w}_t$.

**Fifth Step (Change memory address):** Once both the victim block $\mathbf{w}_t$ and its substitute block $\mathbf{w}_r$ are identified, replace $\mathbf{w}_t$ with $\mathbf{w}_r$ and record their addresses for future references. The search moves to the next iteration and continues until the attack objective is satisfied. At run-time, the attacker launches the fault injection targeting the previously recorded addresses.

## 5 EXPERIMENTAL SETUP

We tested the efficiency of our attack (VLM-PTA) across a wide range of architectures, tasks, and datasets. To evaluate the efficacy of VLM-PTA, we report the number of attack iterations, defined as the total number of altered weight blocks. This metric more appropriately reflects our attack complexity, as prior work (Yao et al., 2020) shows that flipping a single bit or multiple bits within the same address page does not add any overhead. Details of the experiments, including dataset, model, hyperparameters are in Appendix D. In addition, more details of the system setting we used in our evaluation can be found in Appendix D, and the code will be released with final version.

Table 1: *Evaluation of VLM-PTA-U.* Attack Iterations (AI) shows the number of replacements in weight blocks. *PAR refers to Post Attack Recall@1 and PAB refers to Post Attack BLUE-1.* Attack goals for PAA and PAB to decrease Recall@1 and BLUE-1 to zero respectively.

| Dataset | CLIP | | | SigLIP | | | BLIP | | |
|---------|----------|-----|------|----------|-----|------|--------|-----|--------|
| | Recall@1 | AI | PAR | Recall@1 | AI | PAR | BLEU-1 | AI | PAB |
| Flickr8k | 65.53 | 1 | 0.30 | 80.17 | 3 | 0.0 | 59.61 | 14 | 1.325 |
| COCO | 50.0 | 1 | 0.02 | 65.34 | 5 | 0.76 | 56.93 | 12 | 0.0245 |

Table 2: *Evaluation of VLM-PTA-T:* Attack Iterations (referred as AI) shows the number of replacements in weight blocks. Here, ASR refers to Attack Success Rate (%).

| CLIP | | | SigLIP | | |
|-------------|----|------|-------------|----|------|
| Initial ASR | AI | ASR | Initial ASR | AI | ASR |
| 0.0 | 3 | 82.7 | 0.0 | 7 | 94.6 |

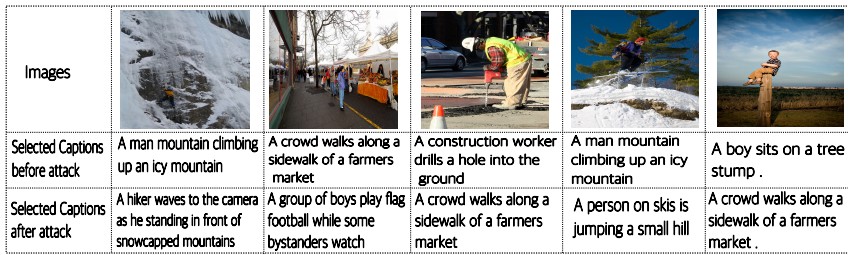

Figure 5: Example of VLM-PTA-U generating wrong caption for each images.

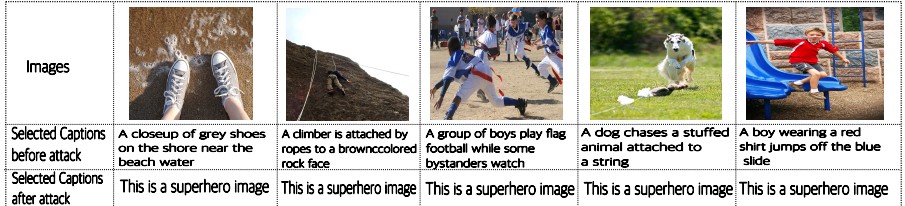

Figure 6: Example of VLM-PTA-T generating a targeted caption for each images.

## 6 EVALUATION OF VLM-PTA

**Evaluation of VLM-PTA-U.** Table 1 summarizes the results of VLM-PTA-U on three VLMs across two datasets, showing the number of iterations required to reduce retrieval (Recall@1) and generative (BLEU-1) metrics to near zero, where the model will retrieve the least relevant or generate random captions as shown in Figure 5. The result in Table 1 shows that VLM-PTA-U can effectively deplete the performance of all three models within 14 attack iterations. In particular, CLIP and SigLIP are more vulnerable to the proposed VLM-PTA-U, requiring less than five attack rounds to compromise the performance.

**Evaluation of VLM-PTA-T.** Table 2 summarizes the results of VLM-PTA-T on VLMs. The Attack Success Rate indicates the percentage of images in the validation dataset that successfully retrieved the targeted captions injected by the attacker. For a targeted attack, we run the attack until the attack success rate stalls and no longer improves for two successive iterations. A qualitative example of retrieved captions on some images before and after the targeted attack is shown in Figure 6. The reason targeted attacks are difficult to use with our attack is that altering weights in a group to achieve gradient descent (for targeted attack) is often challenging. Our evaluation shows we could achieve close to 82-94 % ASR using the proposed VLM-PTA-T.

**Comparison with Competitive Methods.** Although VLM-PTA employs a novel fault-injection mechanism to perturb weights, we compare our algorithm's impact with competitive adversarial weight perturbation attacks (e.g., BFA / T-BFA (Rakin et al., 2019b; 2021a)), which use bit-flips to corrupt model weights directly. *VLM-PTA outperforms BFA by degrading the VLMs caption generation capabilities by a factor of $200\times$, demonstrating superior efficiency and efficacy.* In contrast, BFA fails to degrade models Recall@1 below 21.61 even after 200 rounds of attack iterations, reflecting proposed VLM-PTA is the most successful adversarial weight pertubation attack to date.

Table 3: Comparing with competitive weight pertubation attack, e.g., bit-flip attacks (BFAs) (Rakin et al., 2019b; 2021a) and VLM-PTA-U on VLMs (CLIP on flickr8k). The attack goal is to decrease Recall@1 and BLUE-1 to zero. *PAR refers to Post Attack Recall@1 and PAB refers to Post Attack BLUE-1, and AI refers to Attack Iterations.*

| Method | CLIP | | | SigLIP | | | BLIP | | |
|--------|----------|-----|-------|----------|-----|-------|--------|-----|-------|
| | Recall@1 | AI | PAR | Recall@1 | AI | PAR | BLEU-1 | AI | PAB |
| BFA | 65.53 | 200 | 21.61 | 80.17 | 200 | 51.32 | 59.61 | 200 | 59.51 |
| VLM-PTA | 65.53 | 1 | 0.30 | 80.17 | 3 | 0.0 | 59.61 | 14 | 1.325 |

Table 4: Energy and latency comparison between VLM-PTA, BFA, T-BFA on CLIP trained on flicker8k. The Energy and Latency are reported in mJ and ms respectively.

| Untargeted Attack | | | | Targeted Attack | | | |
|--------|------------|-------------|--------------|-----------|------------|-------------|--------------|
| Method | Iterations | Energy (mJ) | Latency (ms) | Method | Iterations | Energy (mJ) | Latency (ms) |
| BFA | 200 | 6.6 | 110 | T-BFA | 200 | 1.09 | 18.15 |
| VLM-PTA-U | 1 | 0.042 (3.2×) | 0.688 (3.2×) | VLM-PTA-T | 2 | 0.44 (2.49×) | 7.29 (2.49×) |

Such inefficiency is reflected in our evaluation further in Table 4. From the results, the attack overhead of VLM-PTA is significantly lower compared to BFA (3.2×) and T-BFA (2.49×). The detailed computation steps for these energy and latency numbers for DDR4 are provided in Appendix D. This reduction in energy and latency is because flipping a single bit or multiple bits within the same address row page does not substantially impact the attack overhead. Hence, our attack's lower energy and latency costs are attributed to its lower number of attack iterations (200× less).

**VLM-PTA against Prior Defenses.** To the best of our knowledge, there are currently no defenses specifically designed to protect VLMs against adversarial weight attacks. Therefore, we adopt two state-of-the-art defense approaches, FaR (Nazari et al., 2024), designed for transformers, and Quantization (He et al., 2020), a general defense against adversarial weight attacks and adapt them to safeguard VLM components. Our evaluation reveals clear differences between traditional BFA and the proposed VLM-PTA: while BFA requires hundreds of attack iterations, VLM-PTA achieves attack objective with only a few iterations. Notably, both defenses significantly reduce the model's clean Recall@1, underscoring an inherent robustness–accuracy trade-off. Overall, the results show that *existing defenses are insufficient to protect VLMs against VLM-PTA*, highlighting the urgent need for new, VLM-specific defense mechanisms against proposed VLM-PTA.

**Ablation Study.** We have performed a comprehensive ablation study in Appendix A exhibiting: i) The impact of weight block size , ii) Analyzing the impact of the constraint in equation 3 and 4 and iii) Impact of attacking only Vision Encoder or Only Text Encoder or Both on attack effectiveness. These ablation studies further clarifies our design choices and their impact on attack overhead and effectiveness.

Table 5: *Evaluation of defenses (FaR and Quantization) against BFA and VLM-PTA attacks on CLIP with Flickr8k.* PAR refers to Post-Attack Recall@1. Attack goal is to decrease Recall@1 to zero.

| Defenses | BFA Attack | | | VLM-PTA Attack | | |
|----------|------------|-----|------|----------------|-----|------|
| | Recall@1 | AI | PAR | Recall@1 | AI | PAR |
| FaR (Nazari et al., 2024) | 32.74 | 200 | 1.05 | 32.74 | 1 | 0.12 |
| 4-bit Quantization (He et al., 2020) | 55.53 | 200 | 1.36 | 55.53 | 5 | 0.06 |

## 7 CONCLUSION

Vision Language Models (VLMs) are vulnerable to input manipulation attacks according to previous studies. However, in this work, we further expose the security of these models against adversarial weight attacks by utilizing a fault injection technique that flips bits in memory page tables. Our proposed attack *VLM-PTA* addresses the unique challenges associated with the implementation of this fault injection by providing theoretical analysis and proposing an optimization search method (Block-Flip). We demonstrate that the proposed VLM-PTA is the most successful attack to date in terms of effectiveness, cost, and ability to bypass defenses.

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

# A    ABLATION STUDIES

To have better insight into whether directly using the optimized substitute block in place of a weight block within the set of VLM weight blocks is presented in Table 6. Additionally, Table 7 illustrates the effect of varying weight block sizes on attack efficacy. Finally, Table 8 shows the impact of attacking each component of the VLM individually.

Table 6: Performance of VLM-PTA on CLIP (Flickr8K) by directly using the value of optimized substitute block instead of replacing with the most similar weight block within VLM to the optimized substitute replacement block, a constraint that was imposed by equation 3 and 4 to reduce additional write operation. The results clearly demonstrate that the optimization constraint has a negligible impact on attack efficacy.

| Method | VLM-PTA-U | | VLM-PTA-T | |
|---|---|---|---|---|
| | Recall@1 | Attack Iterations | Recall@1 | Attack Iterations |
| Writing the optimized substitute block | 0.370 | 1 | 97.96 | 5 |
| Replace with highly similar block to the optimized block | 0.30 | 1 | 82.7 | 3 |

Table 7: Performance of VLM-PTA on CLIP (Flickr8K), with using different block sizes. The results clearly exhibit that the impact of weight block size is minimal on the un-targeted attack, while the targeted attack performs well with a smaller block size.

| Block size | Untargeted Attack | | Targeted Attack | |
|---|---|---|---|---|
| | Recall@1 | # of iterations | ASR | # of iterations |
| 1 | 0. 0617 | 2 | 99.62 | 4 |
| 128 | 0. 494 | 2 | 74.61 | 7 |
| 512 | 0. 1235 | 1 | 78.13 | 3 |
| 1024 | 0.308 | 1 | 82.70 | 2 |

Table 8: Performance of VLM-PTA on CLIP (Flickr8K) by attacking different components of VLM (vision text/ both encoders). The results exhibit that attacking any component individually yields similar performance using VLM-PTA. However, utilizing the Vision component makes BFA slightly more potent.

| Encoder | VLM-PTA | | BFA | |
|---|---|---|---|---|
| | # of iterations | Recall@1 | # of iterations | Recall@1 |
| Vision | 3 | 0.3088 | 200 | 1.852 |
| Text | 4 | 0.802 | 200 | 7.411 |
| Vision and Text | 1 | 0.300 | 200 | 21.61 |

# B    EXTENDED RESULTS BEYOND VLM

Our attack was motivated to design and target VLM, considering the shortcomings of other attacks in the VLM domain. However, Table 9, 10, and 11 show the performance of VLM-PTA-U and VLM-PTA-T across different DNN architectures on three benchmark datasets: CIFAR-10, CIFAR-100, and ImageNet. The results demonstrate that the VLM-PTA-U attack can degrade model performance to the level of random guessing in fewer than 10 iterations, while the VLM-PTA-T attack successfully enforces classification into the target class within a maximum of 29 iterations.

Table 10: Performance Summary of VLM-PTA-U and VLM-PTA-T Attacks on CIFAR-100 dataset.

| Model | Untargeted Attack | | Targeted Attack | |
|---|---|---|---|---|
| | Initial ACC (%) | Iterations | Initial ASR (%) | Iterations |
| ResNet-20 | 62.82 | 4 | 1.09 | 16 |
| ResNet-32 | 65.23 | 4 | 0.91 | 2 |
| ResNet-44 | 68.06 | 2 | 0.93 | 2 |
| ResNet-56 | 67.36 | 2 | 0.82 | 2 |
| MobileNetV2 | 64.80 | 1 | 0.74 | 22 |
| ShuffleNetV2 | 67.08 | 3 | 0.79 | 29 |

Table 9: Performance Summary of VLM-PTA-U and VLM-PTA-T attacks on CIFAR-10 dataset.

| Model | VLM-PTA-U Attack | | VLM-PTA-T Attack | |
|---|---|---|---|---|
| | Initial ACC (%) | Iterations | Initial ASR (%) | Iterations |
| ResNet-20 | 92.40 | 7 | 9.95 | 4 |
| ResNet-32 | 93.45 | 4 | 9.79 | 8 |
| ResNet-44 | 93.90 | 10 | 9.95 | 7 |
| ResNet-56 | 94.28 | 4 | 9.81 | 5 |
| MobileNetV2 | 93.00 | 2 | 9.75 | 16 |
| ShuffleNetV2 | 93.25 | 5 | 10.01 | 6 |

Table 11: Performance Summary of VLM-PTA-U and VLM-PTA-T Attacks on ImageNet dataset.

| Model | Untargeted Attack | | Targeted Attack | |
|---|---|---|---|---|
| | Initial ACC (%) | Iterations | Initial ASR (%) | Iterations |
| ResNet18 | 69.498 | 5 | 0.098 | 9 |
| ResNet34 | 73.126 | 5 | 0.100 | 15 |
| ResNet50 | 75.834 | 2 | 0.092 | 10 |
| DenseNet121 | 74.248 | 9 | 0.096 | 8 |
| DenseNet169 | 75.358 | 3 | 0.096 | 5 |
| DeiT-S | 79.644 | 9 | 0.086 | 10 |

## C  DETAILED THREAT MODEL

Our attack adopts a standard practical threat model following the attacker privileges established by prior works, including both system (Yao et al., 2020; Rakin et al., 2022; Hong et al., 2019; Zhang et al., 2020) and software-level (Rakin et al., 2019b; Chen et al., 2021; Rakin et al., 2021b) exploitation. For the system level, we assume the VLM inference is running on a resource-sharing environment, which is practical due to the recent popularity of Machine-Learning-as-a-Service (MLaaS) (Ribeiro et al., 2015). The attacker can run user-level unprivileged processes remotely on the same machine as the victim's VLM. The attacker can map the virtual addresses to physical addresses using several techniques such as leveraging huge page support, hardware-based side channel attack (Gruss et al., 2018), and memory messaging (Kwong et al., 2020). The attacker requires knowledge of the DRAM memory addressing scheme, which can be obtained via reverse-engineering (Pessl et al., 2016). We assume the attacker can cause a targeted bit-flip to the page table and cause a bit-flip at the desired location using fast and precise multi-bit-flip techniques (Yao et al., 2020), which includes entire address bit profiling, then fast and precise bit-flipping using rowhammer. We use double-sided rowhammer where an attacker can set specific bit patterns (Rakin et al., 2022) in the aggressor rows to achieve targeted bit-flips. Following the existing rowhammer attack setting (Kim et al., 2014; Yao et al., 2020; Zhang et al., 2020), we will utilize read disturbance as a mechanism to induce a fault in the victim page table only. While prior attacks exploit memory data corruption directly, our novel attack perspective is to exploit the memory address space through side-channel only. Nevertheless, we assume the kernel and operating system are trusted and well-protected (Konoth et al., 2018). Again, following standard practice, we assume the commercial DRAM is not protected by ECC and thus cannot protect large-scale VLM against rowhammer (Yao et al., 2020; Rakin et al., 2022).

For software VLM, we assume a white-box attacker, following the standard threat model of prior adversarial weight attack (Rakin et al., 2019b; Hong et al., 2019; Yao et al., 2020; Chen et al., 2021; Rakin et al., 2021b). In a white-box threat model, the attacker can access model weights, architecture, and some sample test data. The recent advancement of side channel attacks to extract the black-box model makes the white-box assumption more practical. Prior works have demonstrated that an attacker can effectively steal layer number, layer size, weight bit size, and parameter values through remote side channel attacks (Yan et al., 2020; Xiang et al., 2020; Yu et al., 2020; Rakin et al., 2022). However, even for a white-box threat model the attacker cannot access training information (i.e., training dataset, hyper-parameters). An attacker can only access the inference stage model and flip (0 to 1 or 1 to 0) identified bits of memory addresses. *In summary, our threat model follows conventional threat model for adversarial weight attack established in the literature (Hong et al., 2019; Yao et al., 2020; Rakin et al., 2019a; Chen et al., 2021).*

## D EXPERIMENTAL DETAILS

**Datasets and Models.** In this work, we evaluate our proposed attack across diverse datasets and architectures, spanning VLMs, deep neural networks (DNNs), and speech models. For VLMs, we use Flickr8k (Hodosh et al., 2013) and COCO (Lin et al., 2014), both consisting of images paired with five captions each, and evaluate three architectures: CLIP (Radford et al., 2021), SigLIP (Zhai et al., 2023) for caption retrieval, and BLIP (Li et al., 2022) for caption generation. For DNNs, we evaluate models on classification datasets with varying numbers of classes and image resolutions: CIFAR-10 (Krizhevsky et al., 2009) (32×32, 10 classes), CIFAR-100 (Krizhevsky et al.) (32×32, 100 classes), and ImageNet (Deng et al., 2009) (224×224, 1,000 classes). On the CIFAR datasets, we test six architectures: ResNet-20/32/44/56 (He et al., 2015), MobileNetV2 (Sandler et al., 2018), and ShuffleNetV2 (Ma et al., 2018). On ImageNet, we evaluate larger models, including ResNet-18/34/50 (He et al., 2015), DenseNet121/169, and the Vision Transformer (DeiT-S).

**Evaluation Metrics and Hyper-parameters.** For untargeted attacks, in VLMs we report the number of iterations required to reduce the retrieval metric (for retrieval models) or the BLEU score (for generative models) to near zero; in DNNs we report the number of iterations required to degrade classification accuracy to the level of random guessing (e.g., 10% for CIFAR-10). For targeted attacks, we report the number of iterations required to increase the Attack Success Rate (ASR) of retrieving the attacker-chosen target caption (e.g., "This is a superhero image") in VLMs; for DNNs, we report the iterations required to force inputs to be misclassified into a specified target class (e.g., $\mathbf{y}_t = 2$).

**Details of System Evaluation Settings** All experiments are conducted in a hardware environment consisting of an Intel Core-i9-10900X, 3.70 GHz CPU with 64 GB RAM, and an Nvidia GeForce GTX RTX A5000 GPU with 24 GB Memory. Python is used to write all necessary codes, and the PyTorch deep learning library is used to implement the VLMs and neural networks. In addition, we experimentally implemented and tested the VLM-PTA on 16 DRAM modules from two major DRAM manufacturers (Hynix and Samsung-16GB, 2400 MHz DDR4) with various die densities and die revisions (as listed in Table 12) by modifying the FPGA-based testing infrastructure in (Olgun et al., 2023) to understand the attack behavior. Our testing infrastructure shown in

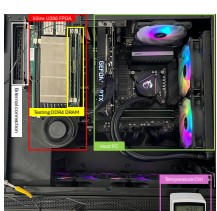

Figure 7: Our testing infrastructure for DDR4 modules.

Figure 7 consists of the Alveo U200 Data Center Accelerator Card (Ale, 2021) as the FPGA that accepts DDR4 modules and runs the test programs by sending DDR4 command traces generated by the host machine. Besides, to have a fair comparison among under-test DRAM chips, the temperature is kept below 30°C with INKBIRDPLUS 1800W temperature controller. Based on this infrastructure, we test our software attack searching algorithm on the PyTorch platform similar to the experimental evaluation platform developed by prior work (Yao et al., 2020) for rowhammer attack evaluation. Following the address flipping attack setting in (Saxena et al.), we assume an OS page size of 4KB and our 8-bit evaluation model. We will identify a specific address (A2) shown in Figure 2 that points to one of the weight blocks following the above setting. Then, in the DRAM, we keep opening and closing the adjacent rows of the original page table (A1) until bit-flips occur and make it point to A2. By repeatedly accessing the rows adjacent to the row containing the victim address (A1), we induce electrical disturbances that can cause bit-flips in the victim row. We ensure precise bit-flip at the targeted victim row by ensuring complementary data content in the attacker's aggressor row. These bit-flips can alter the memory contents, changing the address pointer in the original page table to point to A2. This allows us to manipulate the memory to redirect the address to the specific weight location, demonstrating the rowhammer attack's efficacy under our simulation setup.

**Evaluation of Energy and Latency.** Common $t_{RAS}$ values for DDR4 memory modules range from 36 to 48 $t_{CK}$ (Choi et al., 2020), but these values can vary based on the module's speed rating (e.g., DDR4-2133, DDR4-2400, DDR4-3200, etc.). The duration of a clock cycle for DDR4-2133 memory can be calculated as $t_{CK} = \frac{1}{2133\text{MT/s}}$. If we consider the full sequence of activating a row, accessing data, and then precharging the row, the total time would be 70 $t_{CK}$. Moreover, the energy consumption of a single activation operation in a DDR4-2400 memory module is calculated as $E = V_{DD} \times I_{DD} \times t$. To provide a concrete example, let us assume the following typical values

for DDR4 memory: $V_{DD} = 1.2V, I_{DD} = 30mA$, $t$ is the total time of activating the row. Taking into account the row-to-row delay, the total energy consumption is approximately 1644.6 pJ.

Table 12: Under-test DRAM chips.

| Vendor | #Chips | Freq (MHz) | Die rev. | Org. | Date |
|---|---|---|---|---|---|
| mf-a (SK Hynix 16GB) | 16 | 2400 | A | x8 | 1817 |
| mf-b (Samsung 16GB) | 16 | 2400 | B | x8 | 2053 |

# E LLM USAGE

In this paper, we employed large language model (LLM) to assist with grammar correction, spelling refinement, and contextually appropriate word selection.

