# OpenReview forum: "VLM-PTA: Exploiting Page Table Attack to Deplete the Intelligence of VLMS"
_ICLR.cc/2026/Conference — ICLR 2026 Conference Withdrawn Submission_

### Official Review · Reviewer_KCen · 2025-10-29

**Soundness:** 3
**Presentation:** 2
**Contribution:** 2
**Rating:** 2
**Confidence:** 3

**Summary:**

The paper presents VLM-PTA, an adversarial weight perturbation attack designed to reduce the performance of VLMs.

The threat model considered allows the attacker (1) access to (possibly a copy) of the target model, (2) knowledge of the memory addressing scheme of the model weights, and (3) ability to flip bits in the page frame number.

Broadly speaking, VLM-PTA contains the following steps:

1) Use gradients to identify the most vulnerable weight block.
2) Calculate a substitution to the target weight block that will degrade model performance.
3) Find the substitute weight block as the closest (in terms of dot product) to the ideal substitution found in step (2).
4) Flip bits in the page frame number to execute substitution.

The method is optimized to require few bit flips, and shows favorable performance against other attacks under the same threat model, including a random substitution baseline (Figure 1).

They consider two different losses optimize with the attack. VLM-PTA-U is untargeted, meaning the goal is for the attacked model to select any incorrect caption. VLM-PTA-T is targeted, meaning the goal is for the attacked model to select a specific incorrect target caption.

**Strengths:**

#### Originality and Quality

The attack appears to be original. In particular, the method to specifically optimize the location of weight substitution to specifically maximize the targeted or untargeted objective.

Given the threat model, the results are strong. VLM-PTA outperforms two baseline methods significantly in terms of attack iterations (Figure 1). The qualitative results in Figures 5 and 6 help to demonstrate the attack well.

#### Clarity

The idea of the paper are fairly well conveyed. The threat model is explained in detail in the Appendix,

#### Significance

I have concerns about the significance of the paper (see next section).

**Weaknesses:**

The main weaknesses of the paper are:

1. Significance. It is not clear that the proposed attack is possible in real world scenarios. In particular, the threat model allows the attacker a number of affordances that seem unrealistic. The paper would be improved by more clearly explaining real life scenarios in which an attacker would have the access assumed in the threat model, in particular access to target model weights, and ability to flip bits in the page frame number. If the attack is not realistic, then this should clearly be stated, and there should be discussion around how future works may be able alter the threat model to make it more realistic.
2. Some theoretical aspects of the paper are low in quality.
	1. Implication of Lemma 4.1 seems incorrect, lemma 4.1 implies $w_r^*$ increases the loss, not that it is optimal.
	2. Lemma 4.2 assume completely unrealistic premises. In particular that the weights of the VLM are random, and even worse that the optimized substitute weight blocks are also random.
3. The main advantage of the proposed method over BFA and random substitution attack baseline appears to be that it requires fewer iterations for the attack to succeed. It is not well motivated, however, why this is important. For example, in Table 4, the BFA baseline has a larger latency than VLM-PTA, however is still extremely small at 110ms and 18.15ms. The paper would be improved by better explaining why fewer iterations in this kind of attack is so important.

**Questions:**

My questions center around the weaknesses I raised:

1. Can you explain the threat model better. In particular, is it realistic and what are some real world situations where this can occur.
2. Can you explain why reducing attack iterations is so important.
3. Why is this attack specific to VLMs? It seems with a modification to the attack objective it could be used against LLMs?

---

### Official Review · Reviewer_T7rd · 2025-11-01

**Soundness:** 3
**Presentation:** 3
**Contribution:** 3
**Rating:** 6
**Confidence:** 2

**Summary:**

This paper proposes an adversarial weight perturbation attack on Vision Language Models (VLMs) called VLM-PTA. The idea is to identify the most vulnerable weight blocks in a VLM and swap them with other blocks of weights. The authors conduct extensive experiments on popular VLMs such as CLIP, SigLIP, and BLIP, demonstrating the effectiveness of their proposed attack.

**Strengths:**

- Weight perturbation attacks on VLMs is an interesting topic.
- The paper is well-written and easy to follow in general, but some more background information would be helpful.
- Extensive experiments are conducted to validate the effectiveness of the proposed method.

**Weaknesses:**

- The threat model assumes white-box access to the model weights and architecture, which may not be realistic for many deployed VLMs.
- More background information on weight perturbation attacks would be helpful.
- The scalability of the proposed method to larger models is not discussed.

**Questions:**

The threat model seems quite strong, as it assumes full access to the model weights and architecture. Although the authors claim that "this white-box assumption can be utilized through remote side channel attacks," I didn't find any real-world examples where close-sourced VLMs can be successfully extracted. Could the authors provide more concrete evidence or references to support this claim?

I would like to see more background information on weight perturbation attacks in general. Particularly, I am curious about the attack scenario, as the attacker can even modify the model weights. In this case, it seems the attacker can simply make the parameters random noise to achieve performance degradation. Could the authors clarify the practical scenarios and limitations of weight perturbation attacks?

I would like to know the generalizability of the proposed VLM-PTA attack to larger models. As the method requires pairwise comparisons between the victim block and all other blocks in the model. If the model size increases, the computational cost of the attack will also increase significantly. This is more concerning as larger VLMs are more commonly used in practice. Could the authors discuss the scalability of their method to larger models?

---

### Official Review · Reviewer_Z916 · 2025-11-02

**Soundness:** 3
**Presentation:** 3
**Contribution:** 2
**Rating:** 2
**Confidence:** 3

**Summary:**

The paper considers a known page-table attack (PTA) that leverages fault injection to compromise model weights. It proposes an optimization technique to select target weights in order to improve attack effectiveness and demonstrates the resulting degradation in model performance.

**Strengths:**

The paper clearly demonstrates that PTA-based fault injection can significantly degrade model performance, and provides empirical evidence of this effect.

**Weaknesses:**

- The motivation of the attack is questionable. Fault-injection techniques (for example, Rowhammer) are difficult to execute in practice; if an adversary possessed such capabilities, they would likely pursue higher-value objectives (e.g., key extraction) rather than merely degrading model accuracy.

- The threat of PTA  in compromising model's weights is already known. System owners concerned about such threat can adopt mitigations, such as dedicated hardware or trusted data centers, that effectively prevent fault injection. Because the proposed method still depends on PTA, its improved effectiveness does not, in my view, offer substantially new insights for defending AI models.

**Questions:**

It would be good to suggest how the proposed method helps to provide insight in mitigating PTA.

---

### Official Review · Reviewer_8psa · 2025-11-06

**Soundness:** 2
**Presentation:** 2
**Contribution:** 2
**Rating:** 2
**Confidence:** 3

**Summary:**

This paper proposes an adversarial weight perturbation attack against VLMs. Specifically, the authors leverage page table attacks (memory fault injection techniques) and try to bit flip the page frame number to replace a victim weight block of a VLM with another substitute weight block. Their extensive evaluation demonstrates the effectiveness of the proposed attacks across different model architectures, tasks, and datasets.

**Strengths:**

- Adversarial attacks are combined with hardware attacks
- Theoretical Analysis

**Weaknesses:**

## Regarding the precondition on "normal execution"

From Figure 2 (specifically "Attacker's 1st Virtual Address"), it appears that A1 is a virtual address in the attacker's process. However, the caption of Figure 2 states: "In normal execution, PTE A1 maps to Victim Row (W1 weight block)." How is this possible? How can a virtual address in the attacker's process point to a weight block that belongs to a different (the victim's) process?

## Regarding the intermediate steps of the attack

Assume A1 initially maps to some ordinary physical page in the attacker's process. The attack flow described in Section 4.1 appears to be:

1. The attacker rowhammers the PTE for A1 so that the PFN inside A1's PTE is changed to point to the page frame that contains the PTE of A2.
2. The attacker's process writes to the memory at A1, which actually overwrites the bytes of PTE (A2). The attacker uses this to change A2's PFN to point to a different physical frame.
3. The attacker then uses A2 to read/write the newly pointed physical frame.

Is this an accurate interpretation? If so, how exactly can the attacker discover the location (physical frame) that contains PTE(A2)?

## Regarding the relation to the victim's weights W1 and W2

If steps 1–3 succeed and the attacker can read/write arbitrary physical frames via manipulated PTEs, how does that translate into "replacing W1 with W2"? Specifically, both W1 and W2 are located inside the victim process; how would the attacker know the PFNs for W1 and W2 at runtime?

Even if the PFNs for W1 and W2 are known, how does the attacker perform the replacement W1 <- W2 in practice? The paper does not describe the exact mechanism for locating and swapping these PFNs at runtime.

## Regarding the evaluation

The evaluation lacks an end-to-end demonstration of rowhammer on a real, multi-process system. In contrast, the related work "Depleting the intelligence of deep neural networks through targeted chain of bit flips" (DeepHammer, USENIX Security 2020) includes an end-to-end evaluation with real flip rates, concrete hardware fault cases, and timing statistics, that support practical applicability. Relying primarily on simulation, while useful, weakens the realism of the threat model unless the authors provide additional justification and empirical evidence that the required mapping, discovery, and PFN-edit steps are achievable on real systems.

**Questions:**

- How can a virtual address in the attacker's process point to a weight block that belongs to a different (the victim's) process?
- how exactly can the attacker discover the location (physical frame) that contains PTE(A2)?
- both W1 and W2 are located inside the victim process; how would the attacker know the PFNs for W1 and W2 at runtime?
- even if the PFNs for W1 and W2 are known, how does the attacker perform the replacement W1 <- W2 in practice?

---

### Note · Authors · 2025-11-12

I have read and agree with the venue's withdrawal policy on behalf of myself and my co-authors.